# Exploring the Intricacies of Neurogenic Niches: Unraveling the Anatomy and Neural Microenvironments

**DOI:** 10.3390/biom14030335

**Published:** 2024-03-12

**Authors:** Ismael Sánchez-Gomar, Noelia Geribaldi-Doldán, Celeste Santos-Rosendo, Ciro Sanguino-Caneva, Carlos Carrillo-Chapman, Ornella Fiorillo-Moreno, José Luis Villareal Camacho, Elkin Navarro Quiroz, Cristina Verástegui

**Affiliations:** 1Departamento de Biomedicina, Biotecnología y Salud Pública, Área de Fisiología, Facultad de Medicina, Universidad de Cádiz, 11003 Cadiz, Spain; ismael.sanchez@uca.es; 2Departamento de Anatomía y Embriología Humanas, Facultad de Medicina, Universidad de Cádiz, 11003 Cadiz, Spain; noelia.geribaldi@uca.es (N.G.-D.); cristina.verastegui@uca.es (C.V.); 3Departamento de Biología, Universidad de Cádiz, 11519 Cadiz, Spain; celeste.santos@uca.es; 4Programa de Especialización en Neurología, Facultad de Ciencias de la Salud, Universidad Simón Bolívar, Barranquilla 080001, Colombia; csanguino@unisimon.edu.co (C.S.-C.); carlos.carrilloc@unisimon.edu.co (C.C.-C.); 5Clínica Iberoamerica, Barranquilla 080001, Colombia; ornella.fiorillo11@gmail.com; 6Clínica El Carmen, Barranquilla 080001, Colombia; 7Programa de Medicina, Facultad de Ciencias de la Salud, Universidad Libre, Barranquilla 081007, Colombia; josel.villarrealc@unilibre.edu.co; 8Centro de Investigaciones en Ciencias de la Vida, Facultad de Ciencias Básicas y Biomédicas, Universidad Simón Bolívar, Barranquilla 080001, Colombia; 9Programa de Medicina, Facultad de Ciencias de la Salud, Fundación Universitaria San Martin, Barranquilla 080001, Colombia

**Keywords:** anatomy, nervous system, neurogenic niches, neural stem cells, adult neurogenesis

## Abstract

Neurogenesis is the process of forming new neurons from neural stem cells (NSCs). In adults, this process takes place in specific areas of the brain, known as neurogenic niches. These regions have unique anatomical features that have been studied in animal models and in the human brain; however, there are differences between these models that need to be addressed. The most studied areas are the subventricular zone, the lateral and latero-dorsal walls of the lateral ventricles, and the dentate gyrus of the hippocampus (Hp), which are known as the canonical areas. Other, less-studied niches, such as the hypothalamus, the cerebellum, and the amygdala, are known as non-canonical areas. Anatomy occupies a relevant place in adult neurogenesis, in which the tissue architecture and cellular location are necessities for the interaction and release of diverse molecules that allow this phenomenon. The cell arrangement within the niche and the location of the niche itself are of particular relevance to the state in which the NSCs are found. Consequently, the majority of previous discoveries have been related to pathology. While many studies are based on animal models, discoveries related to neurogenesis in humans have also been made; however, in this case, opinions vary, leading to extensive controversy in recent years. In this review, we address the anatomical characteristics of the different brain regions to better understand their relationships within neurogenesis.

## 1. Introduction

Neurogenesis is the process of forming new neurons from neural stem cells (NSCs). The scientific community is deeply interested in elucidating all the mechanisms involved in this process, especially in view of the latest findings related to the adult brain and the aging process. It is well known that NSCs decreases in the aging brain in most mammals, including humans [1,2,3]. Thanks to several discoveries since neurogenesis in the adult brain was first defined, it is now thought that there are at least two main regions in which neurogenesis occurs in adulthood and in the postnatal brain [4], at least in some species. These specific, studied zones are the subventricular zone (SVZ), a specialized wall within the lateral ventricle, and the subgranular zone (SGZ), located in the dentate gyrus (DG) of the hippocampus [4,5]. In these regions, a complex network of stimuli determines a space where NSCs could exist and form new neuroblasts that could mature and eventually integrate into the pre-existing regions [6,7]. Neurogenesis is defined as the process of activation of NSCs in order to form new neuroblasts; however, during this process, NSCs are able to form progenitor cells that have a higher capacity for division while retaining the ability to form the entire lineage of neural phenotypes, including astrocytes, neurons, and oligodendrocytes [8,9,10]. Within the niche, a balance of symmetrical and asymmetrical divisions mark the direction of NSCs [11]. The differences between the SVZ and SGZ regions have been extensively studied in mice, and it is generally accepted that, in the SVZ, neuroblasts migrate long distances to integrate into the olfactory bulb (OB); however, in the DG, they integrate into the hippocampal network [9,12,13]. Regarding the characteristics of the cells comprising the SVZ and the DG, different cell types have been described. In the SVZ, historically, NSCs are also called B1-type cells. These cells show astrocytic characteristics but also have the ability to form all phenotypes of the neural lineage. These cells can be in a quiescent state and become activated in response to various molecular signals [6,14]. Asymmetric divisions are necessary for B1-type cells to form C-type cells (transit-amplifying cells) that have the capacity to form A-type cells (neuroblasts) [6,9]. These terms have now changed; although, we believe it is correct to name them because of their wide-ranging implications for related discoveries. The importance of resident stem cells in relation to tissue regeneration needs to be further explored, especially when it comes to nerve tissue; in fact, despite the presence of resident NSCs under physiological conditions, it is not sufficient to mention regeneration. The findings in murine models and in humans differ greatly, which is why further studies are needed to distinguish between what derives from pathologies and what occurs under physiological conditions. NSCs can spend long periods in a quiescent (qNSCs) state and can be activated (aNSCs) if there is a specific environment for this, be it physiological or pathological conditions [15]. The NSCs present here are located very close to the lateral ventricles, although they do not contact them directly, but do so by means of prolongations that are surrounded by cerebrospinal fluid (CSF). The cell polarity imparted by the apical complex allows for an enrichment of growth factor receptors along the apical surface of cortical progenitor cells, strongly suggesting that progenitors selectively sample the CSF for diffusible signals that instruct neurogenesis and establish relationships between cells and the vasculature [16,17,18,19,20].

Regarding the NSC state, it is accepted that most of the divisions that NSCs undergo are symmetric divisions of the type that form somewhat more differentiated, but self-renewing cells called neural progenitors (NPCs). These can divide rapidly before acquiring a differentiated phenotype. On the other hand, NSCs can also self-renew in a way that ensures maintenance of the cell pool [21]. Many groups have put their efforts into finding ways to distinguish those cells that are in a quiescent state, especially using single-cell sequencing, so that so-called primed NSCs (pNSCs) have been found that contain specific molecular characteristics compared to quiescent cells (qNSCs). These characteristics range from high ribosomal activity to the expression of some genes [22]. qNSCs are characterized by the expression of *Sox9*, *Id2*, and others such as *Id3*, while aNSCs express *Egfr*, *Ascl1*, *Dlx1*, and *Dxl2* [23].

In the case of the DG, specifically in the SGZ, radial glia-like NSCs (RGLs), also called type 1 cells, form intermediate progenitor cells that are able to form neuroblasts [13]. In the SGZ, neuroblasts migrate tangentially and form immature neurons to eventually migrate radially to the granule cell layer, where they differentiate into mature dentate granule neurons [24,25]. A similar situation occurs in this region, where NSCs with limited self-renewal capacity have been found in young individuals [26,27]. These cells can be activated by different signals, including pathological signals or ageing, contributing to their depletion [28,29]. Recently, using single-cell transcriptome analyes, specific molecular components were revealed to have association owing to ageing, specifically the tyrosine-protein kinase Abl1 [30]. Some studies have investigated the migratory processes of various pathology-related regions, pinpointing specific molecules that determine the direction and type of migration of NSCs and their progeny [31,32].

In this review, we show that several animal models have been used, bearing in mind that neurogenesis is a process that has a certain degree of conservation between species. Thus, although most of the studies have been carried out in murine models [33], other authors have used other animals, such as birds [34] and primates [35]. In humans, however, neurogenesis is a process that is still under intense study and is the subject of much controversy. The first evidence came from the work of Eriksson et al. [36], and other studies have followed which we will discuss in this review [3,37,38]. However, it is important to consider that discoveries related to humans, and even in other higher mammals, have sparked extensive debate that continues to this day.

Although the SVZ and the DG of the hippocampus are the most studied areas in relation to the process of neurogenesis in murine models, other areas have been studied or are under study with often controversial results: for example, the human striatum [39], the adult primates amygdala [40], and also in human amygdala [41], the mice substantia nigra [42,43], and the rat third and fourth ventricles [44]. Another important potential niche is the ventral hypothalamic parenchyma, a zone surrounding the third ventricle, which is related to the energy-balance regulatory system [45]. The locations and functions of the most studied regions are summarized in Table 1.

Pathology has always had a relevant place in discoveries, including those in neurogenesis, where animal studies have been used to understand how this process occurs in the human brain. Here, we analyze the anatomical characteristics of the most studied neurogenic niches and other, less explored, niches in order to elucidate their capacity to contain NSCs and their progeny.

## 2. Neurogenesis in the Subventricular Zone

The SVZ, located in both lateral ventricles, is one of the most studied neurogenic regions in mice. It is a large germinal region (a 3–5 mm layer) containing NSCs and their progeny. This layer is located between the lateral ventricles, the corpus callosum and the corpus striatum [46,47,48]. There are substantial differences between the murine SVZ and the human SVZ. In humans, the SVZ is distributed across four layers, each with specific anatomical features (Figure 1A,B). The first layer, known as the superficial ependymal layer or lamina I, is characterized by its close contact with the ventricular cavity. The second layer, or lamina II, is also known as the hypocellular layer, and contains some astrocytes and some ependymal cells, as well as some astrocytic processes. The lamina III, or astrocytic ribbon, where NSCs are present, and the lamina IV, are transition regions between the other layers and the brain parenchyma of the caudate nucleus [49,50]. Notably, in mice, the astrocytes of the SVZ are located next to the ependymal layer, whereas in humans they are located separately [51]. Furthermore, it is noted that in the human SVZ, astrocytes, NSCs, oligodendrocytes, oligodendrocyte precursor cells, ependymal cells, neuroblasts, and neurons are widely present, but multipotent amplifying transient progenitors are not [52]. In relation to lamina IV, a special mention should be made of the relationship between the blood vessels and the neurogenic process, due to their interaction with NSCs [142]. In general, studies in mice show that the proximity of NSCs to blood vessels is important because it ensures access to oxygen, nutrients, and other signaling molecules that regulate the state of these cells. This will ensure that the environment is conducive to ensuring the proliferation of NSCs if the signals so indicate [143,144,145]. For example, near to the SVZ, the vasculature is characterized by a dense network but with some permeability. NSCs can expand long basal processes that envelop the vessels by expressing laminins and integrins typical of the SVZ [146]. In addition, vasculature is related to the migration of NSCs and its progeny, serving as support scaffolds [31,147]. Related to this microenvironment close to the sanguine vessels, non-neurogenic functions can be attributed to NSCs, but taking into account neuromodulatory and neuroprotective functions. This is due to the secretory functions of NSCs and their consequent systemic function [148]. In addition, OB neurogenesis in mice has been described as an additional process throughout life [149,150], whereas, in adult humans, researchers were unable to find neurogenesis in the OB, nor the addition of new neuroblasts generated from the SVZ to the OB [2,151]. However, neuroblasts from the SVZ can migrate to other brain regions in the human brain, such as the basal ganglia and the cerebral cortex, mostly under pathological conditions [31,39,152].

In 2022, Bitar et al. conducted a study employing a transcriptomic analysis of the human SVZ, revealing dynamic changes in the expression of genes associated with neurogenesis and neuroinflammation across a lifespan. Notably, the observed variations implicate neuroinflammation as a key factor directing neural stem cells into a quiescent state [153]. It is worth noting that this phenomenon had previously been elucidated in murine models by other researchers [154].

One of the most important points when studying SVZ neurogenesis in humans at present is its relationship with some types of malignant tumors. This is the case for glioblastoma (GBM), which is the most frequent tumor with the worst prognosis [57,58]. Currently, treatments for GBM are not effective, although they include surgical resection, radiotherapy, and chemotherapy, such as with temozolomide [59,60]. Considering the cytoarchitecture of the SVZ, the presence of NSCs, and their relationship with the vascular system, glioma stem cells (GSCs) could originate from this neurogenic niche [53,54,55]. In addition, the low oxygen concentration present in the SVZ favors the origin of GSCs and their self-renewal, simulating the parenchyma of this type of tumor [61]. Studies based on genome-wide CRISPR-Cas9 revealed that NSCs and GSCs have a high grade of similarity, such as the presence of the transcriptional factor SOX2 [56]. Several studies have been conducted regarding GBM and its SVZ origin; Lee et al., for example, provide strong molecular and genetic evidence that it is a key part of the anatomical description of the SVZ [62].

## 3. Neurogenesis in the Striatum

Anatomically, the basal ganglia include the striatum, globus pallidus, subthalamic nucleus, and substantia nigra, and the latter two regions can be divided into other specific regions [69].

Figure 1A,B, summarizes the localization of the basal ganglia in the human brain. In the lower parts of each cerebral hemisphere, masses of subcortical gray matter, known as basal nuclei, are located lateral to the thalamus. These include the striatum, which is functionally closely related to the subthalamus of the diencephalon, and the mesencephalic substantia nigra. The striatum comprises the lenticular or lentiform and caudate nuclei, which are separated from each other by fibers of the internal capsule. The lenticular nucleus is further subdivided into the medial globus pallidus and the lateral putamen. Both nuclei are separated from the thalamus by the posterior arm of the internal capsule. Lateral to the putamen, there is a thin sheet of white matter called the external capsule, which separates it from another gray nucleus, the claustrum [70]. The function of the striatum is related to muscle movement and motor control. In addition to the function related to motor control, the striatum is considered to be involved in action selection, decision-making, and also in matters related to learning [155,156,157]. In terms of functionality, a big problem is the delimitations derived from the boundaries of each part or section of the striatum. However, three main domains are considered in terms of functionality: a sensorimotor domain (dorsolateral striatum) that deals with habit-related issues; an associative domain (dorsomedial striatum) that deals with goal-directed behavior; and finally, the limbic domain (up to the ventral striatum) that deals with affective functions [158,159,160,161]. The striatum is the highest subcortical connection zone of the extrapyramidal motor system and contains at least two neuronal populations, one expressing ENK and the dopamine receptor D1 and the other expressing SP and the dopamine receptor D2 [71]. The striatum receives connections from the cortex, the thalamus, and the substantia nigra. Here, we can find different fibers, such as corticostriatal fibers; however, this connection is not reciprocal, there are no fibers from the striatum to the cortex. Most of the fibers are direct fibers, while others cross to the opposite side through the corpus callosum. Thalamostriatal fibers cross from the centromedian nucleus of the thalamus to the caudate and putamen. Impulses from the cerebellum and the reticular formation of the midbrain reach the striatum via this pathway. Finally, nigrostriatal fibers are fibers of dopaminergic neurons that cross the internal capsule and pass from the pallidum to the striatum. Although afferent connections are more abundant, the striatum only sends efferent fibers to the pallidum and substantia nigra [72]. 

Controversy surrounds the study of neurogenesis in the striatum, on the one hand, because of its extensive relationship with regions such as the SVZ and because most findings are based on pathological conditions. On the other hand, the controversy increases if we consider humans.

In the striatal parenchyma, there are cells that have the capacity to acquire characteristics of neural precursor cells capable of generating the entire neural cell lineage. These findings, carried out by Zhang and colleagues, used single-cell transcriptomics and pseudotime trajectories to confirm astrocyte reprogramming [162]. These authors carried out studies, using a mice model, to reprogram resident astrocytes from the striatum. Specifically, they studied the DLX2 factor, considering it sufficient to generate reprogramming towards induced neural progenitors (positive for the ASCL1 marker). However, the specific regions of the location of these resident astrocytes and their transdifferentiation capacity are currently under active study. Some studies are based on demonstrating the heterogeneity of different astrocytes in regions of the striatum, in order to shed light on this question by differentiating between the different molecules present in typical grey matter and white matter astrocytes. Khakh et al. showed that mice grey matter astrocytes in the striatum express the gap junction protein connexin 30 (Cx30), but not white matter astrocytes. Astrocytes in the grey matter of the globus pallidus were found to express high levels of Cx30, while those in the neighboring striatum expressed low levels [163,164]. However, in the developing mouse brain, gap junction proteins connexin 43 are differentially expressed and play important roles in the regulation of neurogenesis. In vivo and in vitro studies on the role of connexin 43 in the regulation of NSC in the SVZ of the adult brain have shown that these proteins are required for the maintenance of NSC quiescence and have been linked to the neurogenic capacities of astrocyte activation in the SVZ related to neovascularization [165]. Recent studies reveal that Cx43 promotes the survival of newborn neurons in the adult mouse hippocampus, whereas Cx30 restricts their survival. Others suggest that the regulation of adult neurogenesis is entirely accomplished by Cx43, whereas Cx30 does not influence the process. The channel functions of Cx43 are required for normal neurogenesis in adult DGs [166]. 

In the case of the substantia nigra, the pars compacta is essential for the generation of the neurotransmitter dopamine, which is closely related to the striatum and its motor implications [122]. Neurogenesis in the substantia nigra will be discussed specifically in Section 6. 

Research conducted by Ernst et al. has revealed the ongoing presence of new interneurons in the human striatum. These interneurons are not present in certain diseases, such as the advanced stages of Huntington’s disease, a rare neurodegenerative disorder characterized by movement disturbances [39]. However, other authors have reported that the generation of new neurons in the striatum is not limited to normal conditions but can also occur after strokes in rats and rabbits [63,64,65]. The primary question addressed in these studies was to determine whether these local striatal neuroblasts actually originate in the subventricular zone (SVZ). Some authors have observed that during early postnatal development, newly formed neurons in the striatum do indeed stem from the SVZ [64,66]. Surprisingly, the newborn neurons found in the striatum are quite similar to OB neurons, specifically GABAergic calretinin-positive (CR) interneurons. These CR interneurons are important in the regulation of neuronal activity in various brain regions. They contribute to the modulation of electrical signals between neurons and play a key role in the regulation of brain excitability. [67]. CR-negative granule cells—like striatal interneurons—are also generated in the SVZ and form the islands of Calleja. Both cell types are increased in the human striatum compared to rodents [67,68]. Moreover, in humans, the striatum plays an important role in neuropsychiatric disorders due to its relationship with dopamine [68]. The complex connectivity with other brain regions places the striatum at the focus of research. In particular, the nucleus of the basal ganglia is connected to the cerebral cortex through several anatomic loops that include projections from the cerebral cortex to the thalamus and vice versa [73]. All these connections link the striatum to several brain functions that can be studied thanks to non-invasive neuroimaging methods [74,75]. The primary issue inherent in Ernst et al.’s study pertains to the employed methodology utilizing 14C. As posited by Duque and Spector, this technique introduces challenges related to artifact contamination, thereby impeding the reliable reproducibility of the results [167].

## 4. Neurogenesis in the Dentate Gyrus of the Hippocampus

Another neurogenic region considered as a canonical niche is the hippocampal DG. The hippocampus is an important region of the limbic system, which is located, in humans, in the medial temporal lobe. The hippocampus is involved in several important functions, such as learning and memory, as well as mood regulation. In fact, it is related to short-term memory and is part of the limbic system. It receives a large number of connections that link it to the encoding of olfactory, visual, auditory, and tactile memory. It is the site of decision-making, as it is part of the circuitry that converts short-term memories into long-term memories, thus providing accurate information [86]. Some authors argue that there are models based on “concept cells”, which can only be applied to humans, giving humans cognitive abilities that cannot be extrapolated to lower animals. In fact, this model is based on the overlapping of episodic memory events, which ultimately allows for the consolidation of long-term memory [94]. The dorsal hippocampus is usually considered to be responsible for memory, while the ventral area is related to emotions. Regarding this, some authors suggest the importance of the consolidation of emotions during sleep, where the hippocampus takes on special relevance. Using murine models and electrophysiological approaches, these authors have associated the REM sleep phase with the transmission of information to other structures, more specifically connecting the ventral hippocampus with other brain regions [95]. 

This complex region shows several differences between humans and other mammals and, in recent years, has become the focus of research. The main debates on human neurogenesis in the hippocampus are based on several contradictory findings, which highlight the need for further study of this exceptional region [3,37,84]. 

To understand the whole process involved in human DG neurogenesis, it is important to review the anatomical features of the hippocampus (Figure 2). The hippocampus consists of an aggregation of gray matter located in the parahippocampal gyrus, which is located in the inferior temporal horn of the lateral ventricles. The hippocampus consists of three general zones: the DG, the hippocampus proper, and the subiculum. The subiculum is a transition zone between the other two regions, forming a sort of C-ring. In addition, the hippocampus is divided into a head, a body, and a tail, which extends from the anterior area where the amygdala can be found near to the posterior part of the splenium of the corpus callosum. The head of the hippocampus is separated inferiorly from the parahippocampal gyrus by the uncal sulcus. The fimbria, crus, body, and the column, are relevant regions too. In addition, the fornix, which surrounds the thalamus, is the main output beam [86]. 

The hippocampus is divided into three areas: the CA1, CA2, and CA3. The DG is especially important in this review because of its involvement in the process of neurogenesis. These regions have been extensively studied in murine models, and the structure of these layers can be seen in Figure 3. Regarding the neurons that make up the hippocampus, within the DG, most of the neurons described are of the primary major glutamatergic type; in fact, granule cells and mossy cells are a specific type of glutamatergic cell. It should be noted that the hippocampus also includes interneurons of the GABAergic type [76]. In most mammals, including primates, the dendritic process of granule cells extends into the molecular layer, and cell bodies are present in the granule cell layer. The hilus, a complex layer between the granule cell layer and CA3, contains granule cell axons. The molecular layer is divided into three regions: the outer molecular layer, the middle molecular layer, and the inner molecular layer. Afferent inputs to the DG come from layer 2 of the entorhinal cortex, while the main outputs come from granule cell fibers projecting to the CA3 [87,88,89]. Stem cells are found specifically in the SGZ of the DG. The neurogenic dynamics in the DG constitute a regulated process involving several molecular signals that determine the environment and fate of the cells located there. The process of neurogenesis in this region includes four distinct phases. The first is a precursor phase, in which cells are activated and divide asymmetrically; these cells are called type 1 cells, which are actually radial glia-like cells. Type 1 cells show projections into the molecular layer and express early progenitor markers, such as nestin and the fibrillary acidic protein GFAP. As a result of this phase, transit-amplifying cells, also called type 2 cells, originate [13,77,78]. As mentioned in the introduction section, these type 2 cells are able to migrate tangentially and can be positive (type 2-B cells) or negative (type 2-A cells) for the early neuronal marker doublecortin (DCX) [78,79]. Type 2-B cells are capable of differentiating into type 3 cells, which are nestin-negative, DCX-positive, and express the polysialylated neuronal cell adhesion molecule (PSA-NCAM) [80]. Once these cells mature, they may express the neuronal marker NeuN and calretinin (CR) [81,82]. The newly generated neurons are able to integrate into the hippocampal cornu ammonis (CA3) region. Although hippocampal neurogenesis in animal models is well studied and shows similarities to the SVZ, the analysis of the number of neurons that eventually integrates into the circuits remains unclear. To elucidate the dynamics of neurogenesis in the adult human brain, researchers have used specific techniques, such as the ^14^C method, which provide an approximation of the kinetics of cell populations [83,84]. These studies revealed that neurons are generated throughout life, with a decline during aging, and also showed that cell turnover rates are similar between mice and humans [84]. Studies conducted by Sorrels et al. in 2018 demonstrated that neurogenesis in the aging hippocampus is undetectable. In this previous study, they showed that newly generated neurons in the DG decrease during early life. Using post-mortem adult human samples, they could not detect any immature neurons in the DG. The same results were obtained using monkeys (*Macaca mulatta*) as another study model [3]. However, one year later, Moreno-Jimenez et al. conducted studies into the human DG focused on Alzheimer’s disease (AD). This team was able to find immature neurons in healthy brains and even in the DG of elderly people (over 90 years old). They also found that the number of neurons was lower in AD patients [37]. Today, this topic is still a matter of intense debate in the scientific community, as methodological or interpretation issues are still under debate [85]. Recently, some authors discussed the multiple benefits of single-cell sequencing to clarify methodological issues regarding hippocampal neurogenesis, but concluded that more studies are needed because of the limitation of samples in the human brain [96]. 

Other studies, such as those carried out by Song et al., looked in depth at neurogenesis in the hippocampus. In 2016, he described the relationship of the neurotransmitter GABA to the regulation of neurogenic processing in the adult hippocampus. They argue that cellular dynamics in this area are largely influenced by pre-existing circuits, taking into account various neurotransmitters characteristic of each anatomical region [97]. Previously, these authors proposed that GABA released from hippocampal interneurons promotes RGL quiescence and its survival [98]. Song’s lab recently delved deeper into the study of NSC quiescence in the mammalian brain, pointing out the importance of maintaining quiescent cells for continued neurogenesis in the area. Using a murine model with deletion of the NKcc1 factor, they are able to maintain cell quiescence in the DG. This factor is a major importer of chloride and is important both in early developmental and adult stages of neurogenesis [99].

Although the SVZ and the DG of the hippocampus are the most studied neurogenic regions, other non-traditional regions, such as the hypothalamus, substantia nigra, amygdala, and the cerebellum are currently under investigation. Therefore, in this review, we analyze the anatomical structures of these regions and the latest discoveries related to neurogenesis in these non-canonical areas.

## 5. Hypothalamic Neurogenesis

In a specific analysis of the anatomical features of the hypothalamus (Figure 4), studies addressed that this brain region is complex and has a wide range of differences compared to other regions. This important region of the brain has been studied since ancient times, as noted by Lechan et al. in their 2000 review [100]. The hypothalamus is a part of the diencephalon, located inferior to the thalamus and directly above the pituitary gland. The hypothalamus extends from the optic chiasm, the lamina terminalis, and the anterior commissure, lying rostral to the cerebral pedunculi and caudal to the interpeduncular fossa. The third ventricle can be found in the midline. In the coronal section, the hypothalamus can be divided into a lateral surface (near the thalamus, subthalamus, and internal capsule), a medial surface (including the wall of the third ventricle), a superior surface (hypothalamic sulcus), and an inferior surface (the floor of the third ventricle) [100]. In adult humans, it is considered an endocrine organ that bases its functionality on the balance of energy and fluid regulation, thermoregulation, sleep and wake states, stress responses, growth, and reproductive behaviors. In addition, the hypothalamus plays roles in emotional and social behaviors, although this function is little studied at present. In fact, the hypothalamus contains implications for several neural functional circuits that include many sensory inputs to ultimately respond to multiple hormonal and metabolic signals [101]. In terms of structure, the median eminence of the hypothalamus is very important for understanding the process of neurogenesis in this region. This region is organized into three zones: the ependymal zone, the internal zone, and the external zone. The first zone, the ependymal zone, is part of the floor of the third ventricle and is especially important because of the presence of a special types of cells known as tanycytes, which have microvilli that extend into the ventricle and have contact with the cerebrospinal fluid (CSF) and also extend a cytoplasmic process into the median eminence, creating a protective barrier in the brain [100,102,103,104,105]. Tanycytes (hypothalamic neurogenic cells) are subdivided into four types: α1, α2, β1, and β2. This subdivision is based on their position, gene expression pattern, neuronal contacts, and neurogenic potential. The tanycytes are located in one of the two neurogenic regions considered in the hypothalamus: the hypothalamic proliferating zone (HPZ), specifically in the lower part of this region, the median eminence, and the lateral walls of the third ventricle at the level of the paraventricular and arcuate nucleus. The tanycytes of the HPZ region express markers and gene expression typical of NSCs, such as nestin, vimentin, *Sox9*, *Notch 1* and *2*, *Hes 1* and *5*, and CD63 [106,107,108]. There is some controversy regarding which type of tanycytes present the most neurogenic features. The α1 tanycytes are present in the ventromedial nucleus, while the α2 tanycytes are present near the arcuate nucleus, both with a long process toward the blood vessels. β1 and β2 tanycytes are found in the lateral part of the infundibular recess and in the floor of the third ventricle, specifically in the median eminence, respectively. These cell types also have contact with blood vessels [91]. Some studies suggest that the tanycytes with neural progenitor characteristics are Fgf10+ cells. Hann et al. (2013) suggested that β-tanycytes are the proliferating cells within the hypothalamus [92]. Another report points to α-tanycytes as the neurogenic cells in this region; Robins et al., in 2013, demonstrated that only α-tanycytes positive for the glial fibrillar protein marker GFAP are capable of forming neurospheres in vitro [93]. These proliferating GFAP cells are similar to SVZ NSCs, and a high number of DCX-positive cells have also been observed near the wall of the third ventricle [109]. 

As mentioned earlier, the hypothalamus is one of the least studied areas that shows neurogenesis, although several researchers have directed their attention here due to the functional involvement of the hypothalamus in metabolic and homeostatic issues. Evidence from recent years suggests that there are NSCs in the hypothalamus capable of generating mature neurons that are able to integrate into pre-existing circuits to complete hypothalamic functions [110,111]. As far as hypothalamic neurogenesis is concerned, all the anatomical subregions of the hypothalamus and even their implications overlap most of the time. This issue may explain why the hypothalamic regulatory system is directly related to the neurogenic process in this brain region [90]. In fact, recent studies have linked hypothalamic neurogenesis to obesity and accelerated ageing. Plakkot et al. point out that increased neuroinflammation and oxidative stress due to senescence may alter NSCs’ function in the hypothalamus, and that obesity may induce accelerated ageing [112]. In addition, the hypothalamus has been described as providing long-range regionalized information to the SVZ and may regulate specific subpopulations of NSCs within this niche. Hypothalamic proopiomelanocortin neurons selectively innervate the anterior ventral SVZ and promote NSC proliferation and the eventual generation of granule neurons. Thus, hunger and satiety processing may regulate adult neurogenesis by modulating activity through a connection between the hypothalamus and SVZ that enables on-demand neurogenesis in response to physiology and environmental cues [113]. Also, the impact of acute and intense stress on neurogenesis and neuroinflammation in the mouse hypothalamus has recently been studied, focusing specifically on the three nuclei, PVN, VMN, and ARC. The researchers found that a single stressor was sufficient to have a significant impact on hypothalamic neurogenesis [114].

The first studies demonstrating the existence of neurogenesis were carried out using 5-bromo-2′-deoxyuridine (BrdU) and also using Cre lines, as in the SVZ and the DG of the hippocampus studies [91,92,93]. Furthermore, in animal models, NSCs and part of their progeny that are present in the neurogenic zone of the hypothalamus express DCX and SOX2 [115,116,117,118,119]. However, in humans there are no in-depth studies on this subject, although neurogenesis has been described in the adult hypothalamus in the arcuate nucleus, median eminence, and ventromedial hypothalamus (VMH) [109]. 

As the hypothalamus has several important functions, the regulation of its neurogenic capacities is currently the focus of research. Zhou et al., in 2020, revealed that hypothalamic radial glia (hRG) and hypothalamic mantle zone radial glia are found in the developing mammalian hypothalamus, and that these cells are capable of producing neurons. This group further noted that the structures and marker genes are conserved in the hypothalamus and are present in both mice and humans [120]. Although tanycytes have been described as radial glia cells with neurogenic capacity, only a small number of neurons can be generated in the postnatal hypothalamus compared to other regions, namely the SVZ and DG of the hippocampus [91,92,93,121]. 

The hypothalamus plays a fundamental role in the regulation of energy homeostasis. Most studies on postnatal hypothalamic neurogenesis have been centered on its function in the regulation of metabolism and body weight [168]. The mediobasal hypothalamus serves as a key regulator of food intake and physical activity. Neurons in the arcuate nucleus (ArcN) and the median eminence (ME) directly respond to signals that govern appetite and satiety. Other regions of the hypothalamus, such as the ventromedial nucleus (VMH) and the dorsomedial nucleus (DMH), also play central roles in the regulation of activity and metabolism [169]. Cells with radial glia characteristics, the tanycytes in the arcuate–median eminence, are especially sensitive to leptin that is related to the felling of satiety [170,171]; in fact, these cells express *Irx3* and *Irx5* which are involved in the regulation of energy homeostasis and obesity [172,173]. Observations related to tanycytes have indicated that dietary factors and diet-regulated cytokines may influence hypothalamic neurogenesis levels. It has been observed that a high-fat diet (HFD) and leptin deficiency affected the rate of loss of BrdU labeling, suggesting continuous neurogenesis in the arcuate nucleus (ArcN), which decreases with age and in obesity [174,175]. Another important function of the hypothalamus is that it plays an essential role in the regulation of reproduction and reproductive function in response to seasonal and hormonal signals, which has a significant impact on hypothalamic neurogenesis [109,176,177,178]. The hypothalamus plays a key role in the regulation of body temperature through a variety of mechanisms. The preoptic area (POA) acts as the central hub of this control, while other hypothalamic regions contribute to temperature-related behaviors. Adult neurogenesis in response to chronic heat exposure, particularly in the POA, leads to heat acclimation, resulting in lower body temperatures and improved adaptation to heat stress [179,180]. Recent research in these rats revealed that long-term voluntary exercise positively influences blood pressure, food intake, body weight and survival rate. This exercise-induced neurogenesis occurs in the arcuate nucleus (ARN) and median eminence (ME). The α2 and β-tanycytes are potential sources of these newborn neurons, and exercise causes an increase in FGF-2 and EGF expression in the ventricular zone, which probably stimulates tanycyte proliferation, especially after stroke [181]. 

Today, researchers are interested in hypothalamic neurogenesis; however, most of the process remains unknown and requires further study.

## 6. Neurogenesis in the Substantia Nigra

Regarding the neurogenic process, there are other brain regions that are possible candidates to present this phenomenon in the postnatal stage. However, most of the studies have been conducted in animal models, not humans. This is the case for the substantia nigra (SN), a midbrain structure that is divided into two regions: the pars compacta and the pars reticulata (Figure 5). This structure is related to the subthalamic nucleus, the amygdala, the cortex, the habenula, and the basal ganglia [123,124], as we mentioned in Section 4 above. With respect to function, it should be noted, using transgenic mice models, that GABAergic neurons in the pars reticulata inhibit thalamic neurons as the main output of the basal ganglia, but it is also composed of dopaminergic neurons in the pars compacta that modulate thalamic excitability [182]. 

This region is especially interesting due to its relationship with dopaminergic neurons that are lost in diseases such as Parkinson’s disease [43,125]. Some studies in murine models tend to report neurogenesis in this region, but they are generally contradictory, and concrete conclusions cannot be drawn. Some authors point out that neurogenesis in the SN can be demonstrated, since NPCs obtained from that region can become neurons if they are transplanted to a suitable niche [126]. Others state that the neurogenesis of dopaminergic neurons occurs throughout life [183], and that this can be increased if an injury occurs [42]. However, doubts arose when Frielingsdorf et al. reported the absence of dopaminergic neurogenesis; they point out that neurogenesis is an exceptionally rare event that is not easily detectable with standard methods and only under extremely well-defined and specific conditions that are difficult to reproduce in a laboratory [184]. Following this thesis, Arias-Carrión et al. in 2009 critically examined several aspects of SN neurogenesis and observed how numerous studies have provided compelling evidence for the existence of NSCs in the neurogenic regions of the adult brain. These NSCs have the ability to generate new neurons, including those of the dopaminergic phenotype. Interestingly, these NSCs are not confined solely to brain regions conventionally acknowledged as conducive to constitutive adult neurogenesis, but are also distributed in more extensive brain areas, notably including the substantia nigra pars compacta (SNc), historically characterized as primarily non-neurogenic. Furthermore, they describe that despite the presence of precursor cells demonstrating neurogenic potential when cultivated in a controlled cellular environment, the occurrence of in vivo adult nigral neurogenesis appears unlikely under both physiological and pathological conditions, as suggested by Frielingsdorf et al., 2004. Current data do not unequivocally establish that injury to the nigrostriatal dopaminergic projection is, by itself, sufficient to initiate adult dopaminergic neurogenesis in the SNc. Subsequent investigations must address the fundamental question of whether manipulations of the local microenvironment signals within the adult SNc will unleash the latent potential of endogenous stem or precursor cells to replace lost neurons in an anatomically and functionally appropriate manner [125]. Also, Mourtzi et al. conducted studies to demonstrate neurogenesis in the SN under both physiological conditions and conditions of dopaminergic neuron loss. They found an extended neurogenic system in the SN with potential to generate dopaminergic neurons, using a model of Parkinson’s Disease. One of the strongest pieces of evidence of this study is the ability of SN-derived cells to form neurospheres, but they also point out that these cells are partially derived from the subependymal zone, following the maturation pathway of mature SN dopaminergic cells [43]. 

## 7. Neurogenesis in the Amygdala

The amygdala is one of the relevant structures of the limbic system that is related to emotions, memory, and decision-making. Anatomically, the human amygdala is composed of 13 nuclei located in the rostral temporal lobe [127,128]. The anatomical location of the amygdala refers to the fact that it adjoins the anterior border of the hippocampal formation and the region comprising the lateral area of the inferior horn of the lateral ventricles (peri-amygdaloid cortex), which is part of the uncus [129]. As mentioned earlier, the amygdala is related to the hippocampus and its functions, and some studies have reported neurogenesis in this region using primate and murine models [42,130]. Recently, studies carried out by Roeder et al. demonstrated the existence of human amygdala neurogenesis in healthy subjects using lipofuscin quantification and ^14^C-based methods in post-mortem samples. These newly generated neurons are similar in number to those obtained in the human hippocampal DG [131]. The postnatal amygdala increases in volume with respect to birth, with dendritic enlargement, synaptogenesis, and gliogenesis being the most important contributing factors. Some researchers have suggested that other neuronal factors may explain this increase in volume. Avino et al., for example, proposed that immature neurons (bcl-2+) have been identified in the amygdala and that they may be able to migrate and mature in the postnatal stage [132]. Although the above authors have provided consistent data on the presence of neurogenesis in the amygdala, there are some limitations in the human study due to methodological issues such as the available markers for immature neurons, and also, the complexity of the carbon dating approach [131,185]. However, Avino et al. suggest that the neuronal factors contributing to the existence of neurogenesis in the area may be due to the maturation of immature neurons found in the paralaminar nucleus, as noted by other authors [185,186,187,188]; in fact, most of these studies were performed using the same immature neuron marker, bcl-2. Other authors focus on the migration of postnatally generated neurons from other areas such as the cortex [40,189]. These findings are important in the sense that they have been made in primates.

Being part of the limbic system, the amygdala has extensive connections with the hippocampus [190], a well-known canonical neurogenic niche, where such methodological problems also existed, as described above [84]. Furthermore, their proximity in their functions has linked both regions, looking for similarities in the neurogenic process and specifically related to neurodegenerative processes.

Recently, cells positive for the marker DCX, which marks immature neurons, were found in layer II of the cortex and also in the amygdala. These experiments, carried out by Li and colleagues, were performed using postmortem samples and demonstrated the presence of these cells at all ages, but it was noted that they decreased with age [41]. However, other authors show that DCX+ cells in the amygdala neurons are born embryonically, not from postnatal neurogenesis, despite a subset retaining immature molecular and morphological features in adults [191,192]. In this regard, Alderman et al. noted that in mice, specifically, neurons present in the paralaminar nucleus of the amygdala are born embryonically and not in postnatal neurogenesis. They propose that these neurons mature during the juvenile stage and then migrate. This region present in mice has its counterpart in humans, and these maturation and migration processes may be related to behavioral changes [191]. Also, Sorrels et al. in 2019 used post-mortem samples of human amygdalae to show that neurons within the paralaminar nucleus mature until adolescence, transitioning into mature neurons, and can maintain an immature state even at advanced ages [192].

## 8. Neurogenesis in the Cerebellum

Considering that the cerebellum is one of the most important areas for coordination and learning, it is to be expected that neurogenesis plays an important role in its maintenance. Anatomically, the cerebellum is composed of two hemispheres that are separated by a layer of dura mater, which are united by an area called the vermis. The cerebellum is located in the posterior cranial fossa, dorsal to the brainstem, and inferiorly to the occipital lobe [133]. The cerebellum is also located behind the fourth ventricle, and the tentorium separate it from the brain. This structure is composed of three lobes: the anterior, posterior (separated by a V-shaped primary fissure), and flocculunodular (separated from the posterior lobe by the posterolateral fissure) lobes [134,135]. Multiple functions are attributed to the cerebellum, not only motor coordination. Its implications in cognitive processes, emotions, and behavior have been well studied. In addition, it has been associated with other aspects such as ataxia, dystonia, autism, and visual dysfunctions [136]. When analyzing the anatomy of the cerebellum, it can be observed that it consists of an external layer of gray matter that presents convolutions (the cerebellar cortex) and internal white matter (the corpus medullaris); these structures form what is known as the “tree of life.” Embedded within the white matter are a series of deep nuclei, the fastigial nuclei, the interposed nuclei (composed of the globose and emboliform nuclei), and the dentate nuclei [134]. Fibers from the globose, emboliform, and dentate nuclei exit the cerebellum through the superior cerebellar peduncle, while fibers from the fastigial nucleus exit through the inferior cerebellar peduncle [134,137].

Two germinal zones capable of harboring NSCs can be found in the cerebellum: the ventricular zone and the rhombic lip [138]. Like many of the discoveries related to neurogenic zones, they are associated with pathology. Wojcinski et al. demonstrated in perinatal animals that nestin-positive cells were able to give rise to granular precursor cells or granular neurons after injury, but not under normal conditions. These authors also suggested that nestin-expressing progenitors are multipotent and have the capacity to self-renew [139]. In addition, some studies have demonstrated that cells expressing the proliferation marker nestin located in the cerebellum have been linked to the origin of medulloblastomas [140]. Other, more recent studies have revealed that the expansion of nestin-positive progenitors using pharmacological agents contributes to the replacement of damaged or lost cells after cerebellar injury [141]. However, it is clear that more work is needed to demonstrate neurogenesis in this area, especially if it is not associated with lesions.

## 9. Discussion

In the present review article, we have pointed out the different anatomical characteristics of the neurogenic niches. A neurogenic niche can be considered an anatomical region within the brain where there is a specific cytoarchitecture and that, together with different signaling signals, generates a specific microenvironment capable of hosting NSC [193,194]. The NSCs in this region are arranged according to a specific organization that ensures the necessary connections in order to regulate the processes of self-renewal and differentiation that ensure homeostasis in the adult brain. Extracellular signals, growth factors, neurotransmitters, and extracellular vesicles contained in the CSF influence the different states in which NSCs can be found. In addition, CSF also aids in the removal of metabolic waste products from the central nervous system, which helps to maintain a suitable environment for neuronal development. [195]. Like other types of stem cell, NSCs can be found in different stages of the cell cycle, being in either a quiescent state for long periods of time or an activated state. They can also produce their progeny slowly but steadily [15,196]. As mentioned above, in the adult brain, the SVZ and the DG of the hippocampus are the most studied niches in terms of neurogenesis. However, this phenomenon has also been studied in other niches that are considered non-canonical [4,5,197]. Several works have studied this process outside the conventional neurogenic niches, but these studies were mostly associated with some types of lesions that serve as stimuli [198]. In general, in these conventional and unconventional regions, there are very specific anatomical characteristics, such as the arrangement of cells in the brain parenchyma or their relationship/proximity to the ventricles and, therefore, to the CSF. These anatomical characteristics mark the differences in the processes and cellular dynamics within the niche and are important in the development of different therapies that can act in these regions in addition to providing the possibility of searching for factors that are potential targets for pharmacological treatments to solve different diseases that involve neuronal loss. Although most of the conclusive studies are based on animal models, there is currently much debate about the existence of neurogenesis in humans. In some studies, the controversy revolves around methodological issues, as can be seen in the studies of Eriksson and Ernst [36,39], but in other cases it is about other problems or interpretations, but ultimately it is at the forefront of current research. Studies performed on post-mortem samples raise the possibility of the existence of neurogenesis in adult individuals and even more so in individuals with pathologies such as AD [37,85]. A question arises here about the interpretation of markers such as DCX, which is present in immature neurons. Some relevant authors suggest that DCX is expressed in non-neurogenic regions of the adult brain and that it can be re-expressed in pathological or ageing processes [199,200]. Although much work remains to be carried out to elucidate what is going on in the human brain, the work is becoming more specific and complex.

Bonfanti and colleagues (2021) conducted a comprehensive review addressing similar issues in their study, where they summarized evidence indicating that a substantial proportion of neurons expressing early markers are not a result of adult neurogenesis but rather stem from their persistence into later stages of adulthood in the brain. In this particular investigation, they focused on the analysis of PSA-NCAM, which are associated with various biological processes ranging from neurogenesis to senescent processes. Despite leaving numerous questions unanswered, their findings unequivocally underscore the role of PSA-NCAM as a modulator of cerebral plasticity [201].

While this review did not specifically address discoveries relating to the cortex, it is pertinent to acknowledge that analogous investigations have been undertaken. In these studies, certain authors have observed congruent issues related to plasticity in murine models, revealing that neurons generated during prenatal stages undergo a progressive maturation process over time [202,203].

Anatomical studies or studies of the characteristics of neurogenic dynamics have been marked by studies in various pathologies. For example, studies carried out to clarify processes that cause neuronal loss, such as cerebral ischemia, have shed light on the dynamics in niches considered to be not as common as the striatum [204]. However, doubt arises here as to whether the cells that can be seen in the striatum originate in this region or come from the SVZ [39,205,206,207]. However, the previous studies point to several differences between humans and animals, due to the type of neurons generated and relative to the migration direction [39]. In the case of areas such as the hypothalamus, studies have grown in recent years due to their implications for metabolism, hormones, and other determinants. In this region, anatomical determinants have been demonstrated in relation to the distribution of NSCs and, in this case, tanycytes [91,93]. Other regions such as the SN and the amygdala are also being investigated [126,130]; however, further studies must be performed to generate conclusive data regarding the anatomy, and also, cell characteristics and markers. In the case of the cerebellum, the data indicate an increase in evidence related to neurogenesis. In this case, the pathology also takes special relevance, and studies have been conducted to determine specific areas capable of harboring NSCs [138].

In all of the above cases it is clear that more studies are needed to determine the interactions between the anatomy and the different molecules that determine the neurogenic niche, including the correlation and extension of existing studies between animals and humans. Also, more in-depth studies are required to determine the possibility of neurogenesis in higher mammals, including humans, with particular emphasis on resolving methodological or result interpretation challenges. Nevertheless, the debate sparked by these investigations is highly promising and intriguing in any case.

This is of vital importance to determine possible therapeutic targets in the long process of curing diseases involving neuronal loss. Many research groups have this goal of exposing the relevance of this topic to today’s scientific development.

## Figures and Tables

**Figure 1 biomolecules-14-00335-f001:**
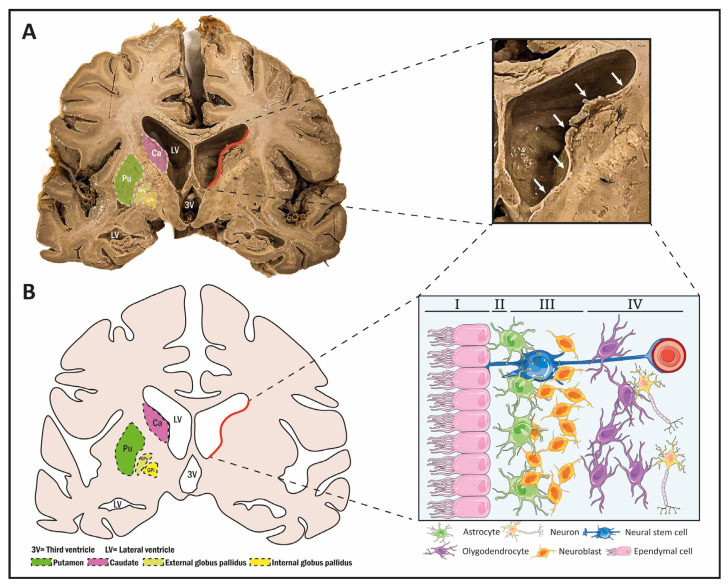
(**A**) Photograph of a coronal view of an adult human brain (genicocapsular section) showing the lateral wall of the lateral ventricle (red line and white arrows), the caudate nucleus (purple), the putamen (in green), and the two globus pallidus nuclei (internal and external, in yellow). (**B**) Drawing depicting a coronal section at genicocapsular level pointing out the lateral wall of the lateral ventricle (red line and white arrows), the caudate nucleus (purple), the putamen (in green), and the two globus pallidus nuclei (internal and external, in yellow). The enlarged region corresponds to the lateral wall of the lateral ventricle where the cytoarchitecture of the region can be observed. Astrocytes are represented in green, neurons in blue, oligodendrocytes in purple, neuroblasts in yellow, and finally ependymal cells are represented in pink.

**Figure 2 biomolecules-14-00335-f002:**
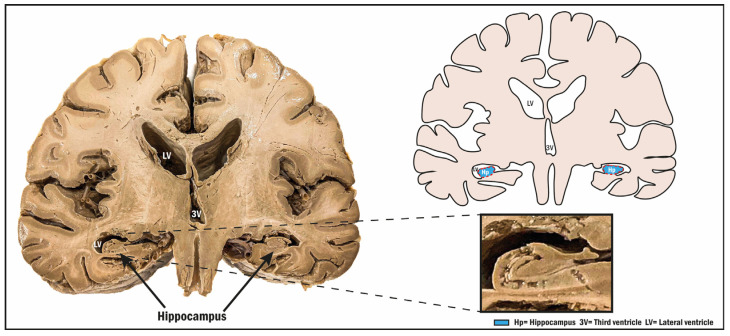
Photograph of a coronal view of an adult human brain (retrolenticular section) showing the anterior and inferior horn of the lateral ventricles, exposing the hippocampus (enlarged region). The drawing shows the anatomical region occupied by the hippocampus.

**Figure 3 biomolecules-14-00335-f003:**
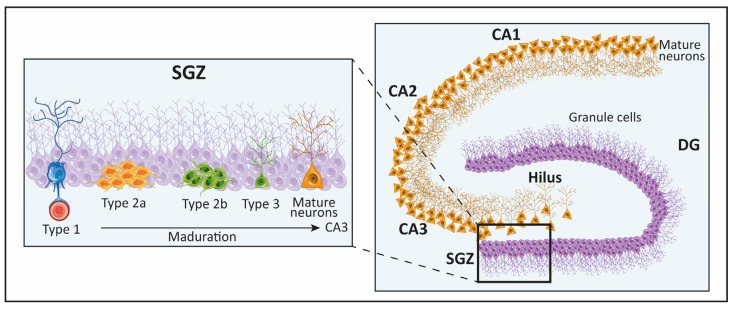
Drawing of the anatomical features of the hippocampus in murine models. The enlarged region corresponds to the dentate gyrus (DG) of the hippocampus, where the proliferating zone corresponds to the subgranular zone (SGZ).

**Figure 4 biomolecules-14-00335-f004:**
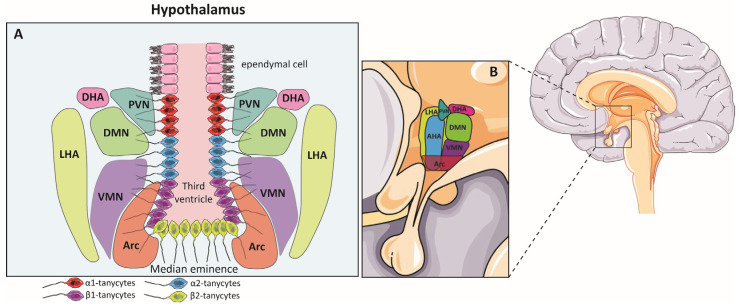
(**A**,**B**), drawings showing the cellular distribution in the hypothalamus and its different regions. Cells can be observed surrounding the third ventricle and distributed in a very specific way. Arc (arcuate nucleus); VMN (ventromedial nucleus); DMN (dorsomedial nucleus); PVN (periventricular nucleus); DHA (dorsal hypothalamic area); LHA (lateral hypothalamic area); AHA (anterior hypothalamic area).

**Figure 5 biomolecules-14-00335-f005:**
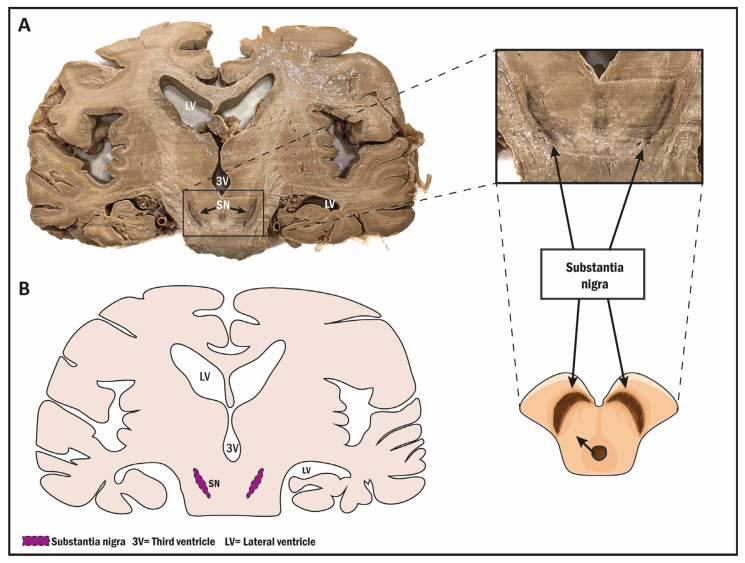
(**A**) Photograph of a coronal view of an adult human brain at the pyramid level, showing the substantia nigra position (black arrows). (**B**) Drawing showing the substantia nigra (purple). 3V (Tirth ventricle); LV (lateral ventricle); SN (substantia nigra).

**Table 1 biomolecules-14-00335-t001:** Most studied region within the human brain regarding the neurogenic process.

Region	Anatomical Localization within the Brain	Function in the Human Brain	References
Subventricular Zone (SVZ)	The lateral and latero-dorsal walls of the lateral ventricles. Between the corpus callosum and the striatum.	Neurogenesis.	Kishi, 1987; Luskin, 1993; Lois and Alvarez-Buylla, 1994; Quinones-Hinojosa et al., 2006; Quinones-Hinojosa et al., 2007; Sanai et al., 2004; Altmann et al., 2019 [46,47,48,49,50,51,52]; Ming and Song, 2011 [4]; Shen et al., 2008; Tavazoie et al., 2008; Sonoda et al., 2014 [53,54,55]; MacLeod et al., 2019 [56].
		Relationship with some types of malignant tumors.	Ostrom et al., 2013; Louis et al., 2016; Stupp et al., 2005; Stupp et al., 2010 [57,58,59,60]; Mohyeldin et al., 2010 [61]; Lee et al., 2018 [62].
Striatum (Str)	Part of the basal ganglia that include the striatum, the globus pallidus, the subthalamic nucleus, and the substantia nigra. Lateral to the thalamus.	Neurogenesis.	Ernst et al., 2014 [39]; Dayer et al., 2005; Luzzati et al., 2006; Yang et al., 2008; Inta et al., 2008; Inta et al., 2015; Inta et al., 2016 [63,64,65,66,67,68].
		Complex connectivity with other regions of the brain. Motor control. Psychiatric disorders.	Alexander and Crutcher, 1990; Yelnik, 2002 [69,70]; Aubert et al., 2000; Lanciego et al., 2012 [71,72].Choi et al., 2012; Di Martino et al., 2008; Barnes et al., 2010 [73,74,75].
Dentate Gyrus (DG) of Hippocampus.	Medial temporal lobe. In the parahippocampal gyrus, inferior temporal horn of the lateral ventricles.	Neurogenesis.	Scharfman, 2016 [76]; Seri et al., 2001 [13]; Filippov et al., 2003; Kempermann et al., 2015; Kronenberg et al., 2003; Seki et al., 2019; Brandt et al., 2003; Kempermann et al., 2003; Spalding et al., 2005 [77,78,79,80,81,82,83]; Spalding et al., 2013 [84]; Sorrells et al., 2018 [3]; Moreno-Jimenez et al., 2019 [37]; Terreros-Roncal et al., 2022 [85].
		One of the parts of the limbic system. Learning, memory, and mood regulation.	Fogwe et al., 2023 [86]; Steward and Scoville, 1976; Amaral et al., 2007; Witter, 2007 [76,87,88,89].
Hypothalamus (HYP)	Diencephalon, ubicated inferior to the thalamus that lies directly above the pituitary gland. Is extended from the optic chiasm, the lamina terminalis, and the anterior commissure, being rostral to the cerebral peduncles and caudal to the interpeduncular fossa. Surrounding the third ventricle.	Neurogenesis.	Kostin et al., 2021 [90]; Lee et al., 2012; Haan et al., 2013; Robins et al., 2013 [91,92,93]; Kokoeva et al., 2005; 2007; Lee and Blackshaw, 2012; Li et al., 2012 [76,77,78,79,80,81,82,83,84,85,86,87,88,89,90,91,92,93,94,95,96,97,98,99,100,101,102,103,104,105,106,107,108,109,110,111,112,113,114,115,116,117,118,119]; Chaker et al., 2016 [115]; Batailler et al., 2014 [109]; Wei et al., 2002; Shimogori et al., 2010; Bolborea and Dale, 2013 [106,107,108]; Zhou et al., 2020; Yoo et al., 2021 [120,121].
		Endocrine organ. Balance of energy and fluid regulation, thermoregulation, sleep and wake states, responses to stress, growth, and reproduction. Emotional and social behaviors. Respond to a multiple hormonal and metabolic signals.	Lechan and Toni, 2000; Burbridge et al., 2016; Monroe, 1967; Knigge and Scott, 1970; Krisch and Leonhardt, 1978; Ugrumov, 1992 [100,101,102,103,104,105].
Substantia Nigra (SN)	Mesencephalon. Related to the basal ganglia, the subthalamic nucleus, the amygdala, the cortex, and the habenula.	Generation of the neurotransmitter dopamine. Motor control. Discussed neurogenesis.	Schultz, 1998 [122]; Lima et al., 2009; Massey and Yousry, 2010 [123,124]; Arias-Carrion et al., 2009 [125]; Mourtzi et al., 2021 [43]. Lie et al., 2002 [126]; Zhao et al., 2003 [42].
Amygdala	Limbic system, rostral temporal lobe. Related with the hippocampus.	Emotions, memory, and decision-making.	Adolphs, 2010; Fox et al., 2015; Rajmohan and Mohandas, 2007 [127,128,129].
		Neurogenesis.	Fowler et al., 2003 [130]; Zhao et al., 2003 [42]; Roeder et al., 2022; Avino et al., 2018 [131,132].
Cerebellum	Posterior cranial fossa, dorsal to the brainstem, and inferiorly to the occipital lobe. Behind the fourth ventricle.	Coordination and learning.	Schmahmann, 1996; Roostaei et al., 2014; Jimsheleishvili and Dididze, 2023; Reeber et al., 2013; Akakin et al., 2014; Millen and Gleeson, 2008; Wojcinski et al., 2017; Li et al., 2013; Jinling et al., 2022 [133,134,135,136,137,138,139,140,141].

Most studied region within the human brain regarding the neurogenic process. The table displays the name of the niche of the region, the localization within the brain, the main functions of the nice, and the references.

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
