# Peer review of "Exploring the Intricacies of Neurogenic Niches: Unraveling the Anatomy and Neural Microenvironments"

_biomolecules, 2024, doi:10.3390/biom14030335_

Round 1

Reviewer 1 Report

Comments and Suggestions for Authors

In the present review with an intriguing title, the authors set the ambition aim to analyze “the anatomical characteristics of the most studied neurogenic niches and other, less explored, niches in order to elucidate their capacity to contain NSCs and their progeny”. However, the authors only repeated well-known facts about the localization of NSC localization and their differentiation potential, which were much better summarized and interpreted in the recently published review by Jurkowski and co-authors (https://doi.org/10.3389/fncel.2020.576444). Moreover, according to the title (Exploring the Intricacies of Neurogenic Niches: Unraveling the Anatomy and Neural Microenvironments), it was expected that the authors would rethink the known microanatomy and at least try to hypothesize how different microenvironments and anatomical structures might affect NSC behavior. In general, the content of the MANU does not reflect either the purpose or the title and does not contain any novelty. To achieve the aim, I would recommend focusing on one canonical and one non-canonical niche, for example the SVZ and the striatum as they border each other, and really try to analyse why neurogenesis is active in the SVZ but not in the striatum; find out which factors from the microenvironment might be involved and how, and conceptualize a hypothesis.

Minor comments

The introduction is too long, chaotic and contains a lot of repetitions.

The figure numbering in incorrect. (there is no Fig. 2.), and the illustrations itself are very poor.

According chapter 2 Anatomy of canonical neurogenic niches 2.1. Neurogenesis in the subventricular zone and the striatum. The striatum is noncanonical neurogenic niche even anatomically borders on SVZ. And neurogenesis in the striatum may consist of two origins, the SVZ and the striatum itself, depending the functional state (for example insult).

There are a lot of incorrect phrases and sentences, for example (line 305) “The hippocampus is involved in several elementary  functions, such as learning and memory, as well as mood regulation.”  - the functions such as learning, memory consolidation, emotion and others could not be characterized as elementary since they are the part of cognition.

Author Response

Reviewer 1

Comments and Suggestions for Authors

In the present review with an intriguing title, the authors set the ambition aim to analyze “the anatomical characteristics of the most studied neurogenic niches and other, less explored, niches in order to elucidate their capacity to contain NSCs and their progeny”. However, the authors only repeated well-known facts about the localization of NSC localization and their differentiation potential, which were much better summarized and interpreted in the recently published review by Jurkowski and co-authors (https://doi.org/10.3389/fncel.2020.576444). Moreover, according to the title (Exploring the Intricacies of Neurogenic Niches: Unraveling the Anatomy and Neural Microenvironments), it was expected that the authors would rethink the known microanatomy and at least try to hypothesize how different microenvironments and anatomical structures might affect NSC behavior. In general, the content of the MANU does not reflect either the purpose or the title and does not contain any novelty. To achieve the aim, I would recommend focusing on one canonical and one non-canonical niche, for example the SVZ and the striatum as they border each other, and really try to analyse why neurogenesis is active in the SVZ but not in the striatum; find out which factors from the microenvironment might be involved and how, and conceptualize a hypothesis.

As the corresponding author, and on behalf of the other authors of the manuscript, we would
like to thank the reviewer for his/her time in reviewing our work. We have carefully evaluated
the reviewer comments and we have tried to reply all questions accordingly. Please, find the
answers to your comments below.

Specific comments:

Minor comments

The introduction is too long, chaotic and contains a lot of repetitions.

We appreciate the reviewer's comment. We have revised the introduction and have trimmed and restructured it to make it more coherent and less repetitive.

The figure numbering in incorrect. (there is no Fig. 2.), and the illustrations itself are very poor.

Thank you very much for the reviewer's suggestion. We have revised the figures and corrected the error. In addition, we have redesigned the figures to make them more interesting and illustrative of our work and have thoroughly revised the numbering and placement in the text.
According chapter 2 Anatomy of canonical neurogenic niches 2.1. Neurogenesis in the subventricular zone and the striatum. The striatum is noncanonical neurogenic niche even anatomically borders on SVZ. And neurogenesis in the striatum may consist of two origins, the SVZ and the striatum itself, depending the functional state (for example insult).

We fully agree and appreciate the reviewer's observation, it was a mistake that we have corrected. Now it has been pointed out every time the findings referred to pathological issues, and moreover we have rearranged the manuscript and the striated body has its own section.

There are a lot of incorrect phrases and sentences, for example (line 305) “The hippocampus is involved in several elementary  functions, such as learning and memory, as well as mood regulation.”  - the functions such as learning, memory consolidation, emotion and others could not be characterized as elementary since they are the part of cognition.

We appreciate the reviewer's comment, we have changed the word "elementary" to "important". We have also checked the entire manuscript for incorrect phrases.

Reviewer 2 Report

Comments and Suggestions for Authors

In their review article titled "Exploring the Intricacies of Neurogenic Niches: Unraveling the 2 Anatomy and Neural Microenvironments" Sánchez-Gomar et al. provide a thoughtful and focused discussion of current and past literature regarding the sites of neurogenesis in the adult brain.  In the course of their review, they specifically highlight anatomical organizational similarities between niche sites of neurogenesis and provide a rationale for the importance of this architecture in the neurogenic process. Additionally, the authors touch on the relevance of specific niches (and their characteristic NSC subtypes) to different physiological and pathological processes. 

Objectively, the review is well-written and properly posed to focus on an important topic that is currently the subject of intense research.  For this reviewer whose expertise is peripheral, the review provided an excellent foundation for the topic of neurogenesis and accomplished the goals of the authors well.  

Comments on the Quality of English Language

The quality of English in the manuscript is acceptable.  Minor issues are 1) formatting errors and inconsistent use of punctuation within several section headings and 2) inconsistent use of abbreviated or written-out forms of brain regions, proteins, and genes.

Examples of these:

1) 2.1. Neurogenesis in the subventricular zone and the striatum 

vs.

2.2 . Neurogenesis in the dentate gyrus of the hippocampus. 

vs. 

3.4 . Neurogenesis in the cerebellum 

Section 2.1. shows the preferred formatting, where the trailing "." is directly after the second digit followed by a space; and no period at the end of the heading. 

Section 2.2 . has a space before the trailing "." and is punctuated

Section 3.1. has the space before the trailing "." but not punctuated. 

2) Doublecortin is defined as "DCX" on line 366 however the spelled-out form is used in a location later in the document

and additional example is "DG" is defined in 46, but dentate gyrus is spelled out in full several times beyond this line

Author Response

Reviewer 2

In their review article titled "Exploring the Intricacies of Neurogenic Niches: Unraveling the 2 Anatomy and Neural Microenvironments" Sánchez-Gomar et al. provide a thoughtful and focused discussion of current and past literature regarding the sites of neurogenesis in the adult brain.  In the course of their review, they specifically highlight anatomical organizational similarities between niche sites of neurogenesis and provide a rationale for the importance of this architecture in the neurogenic process. Additionally, the authors touch on the relevance of specific niches (and their characteristic NSC subtypes) to different physiological and pathological processes.

Objectively, the review is well-written and properly posed to focus on an important topic that is currently the subject of intense research.  For this reviewer whose expertise is peripheral, the review provided an excellent foundation for the topic of neurogenesis and accomplished the goals of the authors well.  

We would like to thank the reviewer for his comments on our manuscript, as he was able to grasp its purpose. 

Reviewer 3 Report

Comments and Suggestions for Authors

The manuscript by Sánchez-Gomar et al., entitled “Exploring the Intricacies of Neurogenic Niches: Unraveling the Anatomy and Neural Microenvironments” is intended to review the organization and extension of the neurogenic niches.

This is a topic that has been reviewed many times by different groups and there is no substantial novelty to justify another review article. In addition, the text suffers of many limitations and mistakes.

- In general, the entire manuscript is too long. It appears as a superficial summary of many studies without focusing on a synthesis aimed at making the point of the current situations. In this mix of studies, results obtained in thousands of papers performed in laboratory rodents (publications that gave us a deep understanding of the canonical brain stem cell niches) are put together with some studies carried out in humans giving contradictory results, some of which are regarded as artifacts or misinterpretations by part of the scientific community. At the end, there is no balanced vision of the current situation, with an overestimation of adult neurogenesis in humans.

Overall, the manuscript is a gathering of information that has already (and better) reviewed in previous reports. It is an old view of the field of adult neurogenesis.

- The physiological aspects of adult neurogenesis and some references to their behavior in pathology are often mixed, without a logical arrangement in the whole text, and without a deep analysis of their relationship.

- Line 41: where the authors write “It is well known that neurogenesis decreases in the aging brain in most mammals, including humans [1]” at least two references must be added (Sanai et al., 2011, Nature; Sorrells et al., 2018, Nature)

- Line 67: where the authors write “The importance of resident stem cells in relation to tissue regeneration needs to be further explored” This has been a dream for many years, yet we now know that the presence of stem cells is not sufficient to obtain neurogenesis, especially in mammals, particularly in humans, due to the non-permissive tissue environment and to evolutionary constraints. In other words, the deep knowledge we now have on the molecular and cellular features of the neural stem cell niches in mice, did not improved our capacity to regenerate the nervous tissue.

- Lines 133-136: where the authors write “Although the SVZ and the DG of the hippocampus are the most studied areas in relation to the process of neurogenesis in murine models, other areas have been described; for example, the human striatum [42], the adult primates amygdala [43], and also in human amygdala [44], the mice substantia nigra [45, 46], and the rat third and fourth ventricles [47]” most of these studies are questionable and/or misinterpreted.

In particular: neurogenesis in the human striatum is claimed on the basis of a single paper performed with C14 and suspected to be an artifact (see Duque and Spector, 2019, Brain Struct Funct); in the primate amygdala there are immature neurons that are not newlyborn (see Chareyron et al., 2022, IJMS); for the human amygdala see Sorrells et al., 2019, Nat Commun.

In brief, that reported here in the manuscript is an old view, that has been changed by new (and ogoing) discoveries.

- Line 152: The subheading “2.1. Neurogenesis in the subventricular zone and the striatum”. On the basis of current knowledge, the subventricular zone and the striatum cannot be put together in a subheading, since we have a huge amount of information on the former (thousands of published papers, mostly performed in mice) and only one claiming it in humans (highly criticized for the suspect of technical artifacts) and only one showing it in rabbits (what is clearly an interspecies difference). Again, the interpretation of data published in mouse (or other isolated animal species) and in humans is not balanced, giving to the reader a biased vision of the state of the art.

- In addition, within the abovementioned subheading there is a mix of data referring to humans and to mice, without clearly stating that results obtained in these species are quite different.

- In addition:

2. Anatomy of canonical neurogenic niches

2.1. Neurogenesis in the subventricular zone and the striatum

In the striatum no niche has ever been described

- In this section, neurogenic processes that have been clearly demonstrated (in the canonical neurogenic niches of rodents) are mixed/listed with other processes that only exist/anatomically demonstrated in some species (e.g., the neurogenesis in the striatum of rabbits) or have been proposed but never confirmed (since they are non-canonical, “incomplete” processes, highly different from those occurring in stem cell niches; see Bonfanti and Peretto, 2011 Eur J Neurosci). This is a major point: the manuscript is focused on stem cell niches but other processes that are not starting from active niches and that are not really neurogenic are mixed in a biased way. There is no clear analysis of these differences (while such analysis has been done in previously published review articles.

- Figure 3. This is another example of biased information. The schematic drawing illustrate the process of genesis and maturation of the granule cells as it has been described in rodents, yet the human brain is showed in the same figure. All this process has not been described in humans and some papers claiming adult neurogenesis in humans have been indicated as artifacts or misinterpretation. Why the reports criticizing the claiming of adult neurogenesis in humans are not cited? (Duque and Spector, 2019, Brain Struct Funct; Duque et al., 2022)

- Lines 673-677: in the Discussion, where the authors write “Although most of the conclusive studies are based on animal models, there is currently much debate about the existence of neurogenesis in humans. This is a methodological issue, as discussed above, and is at the forefront of current research. Many studies performed on post-mortem samples point to the existence of neurogenesis in adult individual…” This biased, for the following reasons:

i) this not only a methodological issue (it is methodological for the Ernst and the Eriksson studies, likely artefacts), but in the case of Moreno-Jimenez et al., 2018, Nat Med, the expression of DCX is true, but it is misinterpreted because there in no niche and no substantial cell division, so these DCX+ neurons are likely non-newly born immature neurons or dematuration (see Hagihara et al., 2019, Mol Brain); i) there are no “many studies” supporting adult neurogenesis in humans, and there are several important papers denying it.

- Line 578: 3.3. Neurogenesis in the amygdala. This is an old view. There is evidence that the DCX+ cells in the amygdala are non-newly generated, rather “immature” neurons that are generated prenatally (see Alderman et al., 2023, Neuron; Sorrells et al., 2019, Nat Commun) Why these important papers are not cited?

(see point below)

- In general, it is completely lacking an entire new research field that can explain some previous misunderstandings about the putative neurogenesis in non-canonical sites: that of the non-dividing, immature or “dormant” neurons. These neurons are prenatally-generated and then remain blocked in a state of immaturity (or arrested maturation), yet they express the same markers of the newluborn neurons in the canonical neurogenic niches. These neurons are present in parenchymal (non-neurogenic) regions of the brain, such as the cerebral cortex and the amygdala. Hence, the presence of neurogenesis in the amygdala is an old view, a bias generated when in the past the presence of doublecortin was considered a specific sign of neurogenesis. But this is no more true. See below a series of papers showing this new aspect of neurogenic plasticity:

Bonfanti & Seki, 2021, Cells

Gómez-Climent et al., 2008, Cereb Cortex;

Luzzati et al., 2009, Cereb Cortex;

Bonfanti & Nacher, 201, Prog Neurobiol2;

La Rosa et al., 2020, eLife

Rotheneichner et al., 2018, Cereb Cortex;

Benedetti et al., 2020, Cereb Cortex

Alderman et al., 2023, Neuron;

Sorrells et al., 2019, Nat Commun

Author Response

Reviewer 3

The manuscript by Sánchez-Gomar et al., entitled “Exploring the Intricacies of Neurogenic Niches: Unraveling the Anatomy and Neural Microenvironments” is intended to review the organization and extension of the neurogenic niches.
This is a topic that has been reviewed many times by different groups and there is no substantial novelty to justify another review article. In addition, the text suffers of many limitations and mistakes.

As the corresponding author, and on behalf of the other authors of the manuscript, we would
like to thank the reviewer for his/her time in reviewing our work. We have carefully evaluated
the reviewer comments and we have tried to reply all questions accordingly. Please, find the
answers to your comments below.

- In general, the entire manuscript is too long. It appears as a superficial summary of many studies without focusing on a synthesis aimed at making the point of the current situations. In this mix of studies, results obtained in thousands of papers performed in laboratory rodents (publications that gave us a deep understanding of the canonical brain stem cell niches) are put together with some studies carried out in humans giving contradictory results, some of which are regarded as artifacts or misinterpretations by part of the scientific community. At the end, there is no balanced vision of the current situation, with an overestimation of adult neurogenesis in humans.
Overall, the manuscript is a gathering of information that has already (and better) reviewed in previous reports. It is an old view of the field of adult neurogenesis.
- The physiological aspects of adult neurogenesis and some references to their behavior in pathology are often mixed, without a logical arrangement in the whole text, and without a deep analysis of their relationship.

We appreciate the reviewer's extensive analysis and have followed all his comments. The manuscript is now tidier and with a more descriptive analysis. We know that the analysis often seems superficial, but it was necessary not to be too long, however we have tried to make more points by comparing the findings in the different species, and also by differentiating pathology and physiological conditions. In addition, we have pointed out controversies and added more relevant literature.

- Line 41: where the authors write “It is well known that neurogenesis decreases in the aging brain in most mammals, including humans [1]” at least two references must be added (Sanai et al., 2011, Nature; Sorrells et al., 2018, Nature)

We appreciate the reviewer's comment. We have ensured to include the references you indicated in the revised version of the manuscript. We believe these additions strengthen the content and enhance the overall quality of the paper.

- Line 67: where the authors write “The importance of resident stem cells in relation to tissue regeneration needs to be further explored” This has been a dream for many years, yet we now know that the presence of stem cells is not sufficient to obtain neurogenesis, especially in mammals, particularly in humans, due to the non-permissive tissue environment and to evolutionary constraints. In other words, the deep knowledge we now have on the molecular and cellular features of the neural stem cell niches in mice, did not improved our capacity to regenerate the nervous tissue.

We appreciate the reviewer's suggestion, and we have clarified what was stated in the corresponding section and subsections.

- Lines 133-136: where the authors write “Although the SVZ and the DG of the hippocampus are the most studied areas in relation to the process of neurogenesis in murine models, other areas have been described; for example, the human striatum [42], the adult primates amygdala [43], and also in human amygdala [44], the mice substantia nigra [45, 46], and the rat third and fourth ventricles [47]” most of these studies are questionable and/or misinterpreted.
In particular: neurogenesis in the human striatum is claimed on the basis of a single paper performed with C14 and suspected to be an artifact (see Duque and Spector, 2019, Brain Struct Funct); in the primate amygdala there are immature neurons that are not newlyborn (see Chareyron et al., 2022, IJMS); for the human amygdala see Sorrells et al., 2019, Nat Commun.

We appreciate the reviewer's appreciation, and we have changed the word "described" to "have been studied or are under study with often controversial results." Additionally, we have analyzed the contributions provided in the corresponding sections. We have clarified the methodological problem raised by the studies of Ernst.

In brief, that reported here in the manuscript is an old view, that has been changed by new (and ogoing) discoveries.
- Line 152: The subheading “2.1. Neurogenesis in the subventricular zone and the striatum”. On the basis of current knowledge, the subventricular zone and the striatum cannot be put together in a subheading, since we have a huge amount of information on the former (thousands of published papers, mostly performed in mice) and only one claiming it in humans (highly criticized for the suspect of technical artifacts) and only one showing it in rabbits (what is clearly an interspecies difference). Again, the interpretation of data published in mouse (or other isolated animal species) and in humans is not balanced, giving to the reader a biased vision of the state of the art.

We fully agree, we have corrected the error. We have created separate sections for the SVZ and the spline. In addition, we have added new findings and also made some observations on what has been written that defend the current controversy on the subject.

- In addition, within the abovementioned subheading there is a mix of data referring to humans and to mice, without clearly stating that results obtained in these species are quite different.
- In addition:
2. Anatomy of canonical neurogenic niches
2.1. Neurogenesis in the subventricular zone and the striatum
In the striatum no niche has ever been described
- In this section, neurogenic processes that have been clearly demonstrated (in the canonical neurogenic niches of rodents) are mixed/listed with other processes that only exist/anatomically demonstrated in some species (e.g., the neurogenesis in the striatum of rabbits) or have been proposed but never confirmed (since they are non-canonical, “incomplete” processes, highly different from those occurring in stem cell niches; see Bonfanti and Peretto, 2011 Eur J Neurosci). This is a major point: the manuscript is focused on stem cell niches but other processes that are not starting from active niches and that are not really neurogenic are mixed in a biased way. There is no clear analysis of these differences (while such analysis has been done in previously published review articles.

We appreciate the reviewer's comments and fully agree. The section has been changed. We have reorganized the manuscript by separating the spline from the SVZ. We have clarified each of the processes by separating the findings in humans and other animals.

- Figure 3. This is another example of biased information. The schematic drawing illustrate the process of genesis and maturation of the granule cells as it has been described in rodents, yet the human brain is showed in the same figure. All this process has not been described in humans and some papers claiming adult neurogenesis in humans have been indicated as artifacts or misinterpretation. Why the reports criticizing the claiming of adult neurogenesis in humans are not cited? (Duque and Spector, 2019, Brain Struct Funct; Duque et al., 2022)

We are in total agreement. The figure has been modified and we have divided it in two to avoid confusion and we have clarified whether it is in rodents or humans.

- Lines 673-677: in the Discussion, where the authors write “Although most of the conclusive studies are based on animal models, there is currently much debate about the existence of neurogenesis in humans. This is a methodological issue, as discussed above, and is at the forefront of current research. Many studies performed on post-mortem samples point to the existence of neurogenesis in adult individual…” This biased, for the following reasons:
i) this not only a methodological issue (it is methodological for the Ernst and the Eriksson studies, likely artefacts), but in the case of Moreno-Jimenez et al., 2018, Nat Med, the expression of DCX is true, but it is misinterpreted because there in no niche and no substantial cell division, so these DCX+ neurons are likely non-newly born immature neurons or dematuration (see Hagihara et al., 2019, Mol Brain); i) there are no “many studies” supporting adult neurogenesis in humans, and there are several important papers denying it.

We fully agree with the reviewer, we did not intend to be biased, we have changed this sentence and explained the differences between the studies. We have also removed the word "many".

- Line 578: 3.3. Neurogenesis in the amygdala. This is an old view. There is evidence that the DCX+ cells in the amygdala are non-newly generated, rather “immature” neurons that are generated prenatally (see Alderman et al., 2023, Neuron; Sorrells et al., 2019, Nat Commun) Why these important papers are not cited?

We appreciate the reviewer's comment. We agree that it is necessary to make this clarification regarding neurogenesis in the amygdala. The references mentioned are also included and explained.

(see point below)
- In general, it is completely lacking an entire new research field that can explain some previous misunderstandings about the putative neurogenesis in non-canonical sites: that of the non-dividing, immature or “dormant” neurons. These neurons are prenatally-generated and then remain blocked in a state of immaturity (or arrested maturation), yet they express the same markers of the newluborn neurons in the canonical neurogenic niches. These neurons are present in parenchymal (non-neurogenic) regions of the brain, such as the cerebral cortex and the amygdala. Hence, the presence of neurogenesis in the amygdala is an old view, a bias generated when in the past the presence of doublecortin was considered a specific sign of neurogenesis. But this is no more true. See below a series of papers showing this new aspect of neurogenic plasticity:

Bonfanti & Seki, 2021, Cells
Gómez-Climent et al., 2008, Cereb Cortex;
Luzzati et al., 2009, Cereb Cortex;
Bonfanti & Nacher, 201, Prog Neurobiol2;
La Rosa et al., 2020, eLife
Rotheneichner et al., 2018, Cereb Cortex;
Benedetti et al., 2020, Cereb Cortex
Alderman et al., 2023, Neuron;
Sorrells et al., 2019, Nat Commun

We appreciate the comments and recommendations from the reviewer. We have incorporated a significant portion of the suggested citations after conducting a thorough study on them. We hope that our revision is now more up-to-date, encompassing all the crucial aspects in the studies on neurogenesis.

Round 2

Reviewer 1 Report

Comments and Suggestions for Authors

The authors have reorganised a MANU and it now looks more logical.

Reviewer 3 Report

Comments and Suggestions for Authors

The authors have revised the manuscript as requested. I think it is now available for publication in Biomolecules.

Comments on the Quality of English Language

There are some repetitions. E.g., in the Abstract:

While many studies are based on animal models, in recent years, discoveries related to neurogenesis in humans have also been made; however, in this case, opinions vary, leading to extensive controversy in recent years